Genomic analyses indicate resilience of a commercially and culturally important marine gastropod snail to climate change

Nimbs Matt J. 1 2 matthew.nimbs@dpi.nsw.gov.au
Champion Curtis 1 2
Lobos Simon E. 3 4
Malcolm Hamish A. 5
Miller Adam D. 3 4
Seinor Kate 1
http://orcid.org/0000-0002-3372-5441 Smith Stephen D.A. 1 6
http://orcid.org/0000-0002-7873-0412 Knott Nathan 7
Wheeler David 8
Coleman Melinda A. 1 2
1 National Marine Science Centre, Southern Cross University , Coffs Harbour, New South Wales , Australia
2 NSW Department of Primary Industries, Fisheries, National Marine Science Centre , Coffs Harbour , Australia
3 Deakin Genomics Centre, Deakin University , Geelong, Vic , Australia
4 School of Life and Environmental Sciences, Deakin University , Warrnambool, Vic , Australia
5 NSW Department of Primary Industries, Fisheries Research , Coffs Harbour, NSW , Australia
6 Aquamarine Australia , Mullaway, NSW , Australia
7 NSW Department of Primary Industries, Fisheries Research , Huskisson, NSW , Australia
8 NSW Department of Primary Industries , Orange, NSW , Australia
Thomas Jonathan
Electronic publication date: 2023 Nov 23
Publication date: 2023
Volume: 11
Electronic Location ID: e16498
Received 2023 Jun 30; Accepted 2023 Oct 31
Copyright: © 2023 Nimbs et al.
Copyright year: 2023
Copyright holder: Nimbs et al.
License: This is an open access article distributed under the terms of the Creative Commons Attribution License, which permits unrestricted use, distribution, reproduction and adaptation in any medium and for any purpose provided that it is properly attributed. For attribution, the original author(s), title, publication source (PeerJ) and either DOI or URL of the article must be cited.
License URL: https://creativecommons.org/licenses/by/4.0/

Keywords: Climate adaptation, Genetic differentiation, Panmixia, Genotype x environment

Funding: NSW Marine Estate Management Strategy This study was funded by the NSW Marine Estate Management Strategy. The funders had no role in study design, data collection and analysis, decision to publish, or preparation of the manuscript.

==============================
Genomic vulnerability analyses are being increasingly used to assess the adaptability of species to climate change and provide an opportunity for proactive management of harvested marine species in changing oceans. Southeastern Australia is a climate change hotspot where many marine species are shifting poleward. The turban snail, Turbo militaris is a commercially and culturally harvested marine gastropod snail from eastern Australia. The species has exhibited a climate-driven poleward range shift over the last two decades presenting an ongoing challenge for sustainable fisheries management. We investigate the impact of future climate change on T. militaris using genotype-by-sequencing to project patterns of gene flow and local adaptation across its range under climate change scenarios. A single admixed, and potentially panmictic, demographic unit was revealed with no evidence of genetic subdivision across the species range. Significant genotype associations with heterogeneous habitat features were observed, including associations with sea surface temperature, ocean currents, and nutrients, indicating possible adaptive genetic differentiation. These findings suggest that standing genetic variation may be available for selection to counter future environmental change, assisted by widespread gene flow, high fecundity and short generation time in this species. We discuss the findings of this study in the content of future fisheries management and conservation.

Introduction

Rapid climate change is impacting the physical state of the world’s oceans and directly threatening the structure and function of marine ecosystems. In particular, ocean warming and marine heatwave events, ocean acidification and deoxygenation, changes to ocean currents, and sea-level rise are already impacting many marine ecosystems and associated socioeconomic and cultural values at a global scale (Brierley & Kingsford, 2009; Doney et al., 2011; Orr et al., 2005; Poloczanska et al., 2013, 2016; Scavia et al., 2002). Shifts in the physical ocean climate pose a direct threat to many of the world’s commercial fisheries, causing changes in species distributions and abundances, altering habitats, decoupling critical trophic interactions, and pushing species beyond their physiological limits (Holland et al., 2021; Hollowed et al., 2013; Roessig et al., 2004; Sumaila et al., 2011). These effects are expected to reduce harvestable biomass in many fisheries (Brander, 2013), with studies predicting reductions in catch of up to 40% for tropical fisheries alone under the RCP8.5 scenario (Lam et al., 2020). Projections suggest that climate change will continue to be an ongoing challenge for sustainable fisheries management into the future (Cheung et al., 2010), highlighting the importance of research aimed at understanding the resilience of individual fisheries to climate change effects, and identifying interventions capable of ‘climate-proofing’ vulnerable fisheries (Fankhauser & Schmidt-Traub, 2011; Harte et al., 2019).

Evidence suggests the ability of many marine species to track their thermal niche via migration is likely to be outpaced by rapid climate change (Hiddink et al., 2012; Vranken et al., 2021). This typically applies to less vagile organisms whose persistence will depend more on their ability to adapt to new thermal environments, either through plasticity or genetic evolution (Donelson et al., 2019; Munday et al., 2013; Somero, 2010). Species with wide latitudinal ranges often show genetically based clines across thermal gradients (Berger et al., 2013; Jenkins et al., 2019; Pereira, Sasaki & Burton, 2017), suggesting standing genetic variation in quantitative traits might be available for adaptation to new environments (Barrett & Schluter, 2008; Sasaki et al., 2022). Yet it is anticipated that selection itself may fail to keep pace with rapid climate change, particularly in long-lived organisms (Vranken et al., 2021; Wood et al., 2021). In such cases, gene flow is likely to play a critical role in the adaptation process, particularly when strong biological connections among locally adapted populations from thermal environments are present (Miller et al., 2019; Sork et al., 2010). Consequently, species with high connectivity among locally adapted populations may be deemed less at risk than those species with limited gene flow and poor dispersal capabilities (Ayre & Hughes, 2004; Coleman, 2015). In such cases, strategic intervention measures may be needed to maximise the adaptive capacity of threatened fish stocks, such as augmentation activities that include deliberately introducing genotypes from warm adapted populations to those at risks of maladaptation (Aitken & Whitlock, 2013; Hagedorn et al., 2018; Layton et al., 2020; Hoffmann, Miller & Weeks, 2021). Consequently, the challenge for fisheries managers is understanding when interventions of this nature will be necessary in order to mitigate the risks of climate change.

Population genetics has played a key role in characterising structure and patterns of gene flow among populations, especially in commercially important marine species over the last few decades (Miller et al., 2019; Smith, Francis & McVeagh, 1991; Ward, 2000; van Oppen & Coleman, 2022). This field of research has been revolutionised by modern DNA sequencing technologies that now allows for genome wide estimates of genetic variation, providing unprecedented power for resolving fine scale patterns of genetic structure (Cheng et al., 2021; Milano et al., 2014; Sağlam et al., 2021) and signatures of adaptive genetic differentiation among fishing stocks spanning environmental gradients (Quigley, Bay & van Oppen, 2020; Riquet et al., 2013). Combined, this information can greatly assist fisheries management by providing insights into the availability of standing variation for adaptation to future environmental challenges, and the potential role of gene flow in assisting the adaptation process (Mason et al., 2022; Papa et al., 2020; Valenzuela-Quiñonez, 2016). Importantly, this information can help to identify fish stocks most at risk of maladaptation, and to guide stock augmentation programs aimed at introducing novel genotypes for selection to act upon to help combat future environmental challenges (Bernatchez et al., 2017; Reiss et al., 2009; Waples & Naish, 2009).

Marine gastropod snails support many commercial and recreational fisheries around the world (Dolorosa et al., 2010; Foale & Day, 1997; Leiva & Castilla, 2002), many of which are expected to be susceptible to climate change effects (Ortíz, Arcos-Ortega & Navarro, 2022; Valles-Regino et al., 2022). Specifically, evidence suggests that ocean warming has the potential to suppress larval development, delay gonad maturation, reduce fecundity, stunt growth, and increase susceptibility to disease in many species, while increased ocean acidification is expected to compromise shell production in others (Holland et al., 2021; Leung, Russell & Connell, 2020; Zacherl, Gaines & Lonhart, 2003). However, some marine gastropods have broad latitudinal distributions and show genetically determined clines across thermal gradients, and strong biological connections over vast geographic distances due to long pelagic larval dispersal stages (Kelly & Palumbi, 2010; Miller et al., 2016; Neethling et al., 2008; Villamor, Costantini & Abbiati, 2014). These findings suggest that some species might possess heritable genetic variation for adaptation to future thermal environments, which maybe assisted by extensive gene flow across thermal gradients (Johansson, 2015; Sanford & Kelly, 2010; Zardi et al., 2011). However, information for some marine gastropod groups is still lacking and is urgently needed to inform future management.

Benthic marine gastropods represent approximately 2% of the global marine mollusc fishery (Department, Food and Agriculture Organization of the United Nations. Fisheries, 1997, 2012), with some species having high economic value being targeted in small-scale artisanal fisheries (i.e Haliotis spp., Strombus spp., Busycon spp, and Concholepas spp.). In recent decades commercial landings have grown with wild-stock catch increasing from 75 k metric tonnes (mt) in 1979 to 103 k mt in 1996 (Department, Food and Agriculture Organization of the United Nations. Fisheries, 1997), however, some gastropod fisheries have been identified as being significantly threatened by climate change (Carranza & Matías, 2023; Ramos et al., 2022). Unfortunately, information of the genetic structure and resilience of many commercially important gastropods is lacking, including trochid snails. To date, only a handful of studies have investigated patterns of population genetic structure among trochoid fisheries (Berry et al., 2019; Díaz-Ferguson et al., 2010; Nikula, Spencer & Waters, 2011), and, to our knowledge, no study has investigated patterns of adaptive genetic variation across the entire range of any trochid species using a dataset that includes thousands of genome-wide variants. Turbo militaris Reeve, 1848, is a large intertidal and shallow subtidal turbinid (Family Turbinidae) from south-eastern Australian coast that is traditionally and recreationally harvested for human consumption (Yearsley, Last & Ward, 1999) and supports a 6.6 tonne turban snail commercial fishery in New South Wales (NSW) (NSW Department of Primary Industries, 2019). Like many trochids, T. militaris is showing signs of climate stress, with evidence of a poleward range shift since the turn of the century (Atlas of Living Australia (ALA), 2018; GBIF, 2016, Benkendorff & Przeslawski, 2008). Given that range shifts in commercially important marine species can have socio-economic and management implications (Madin et al., 2012; Bonebrake et al., 2018), it is prudent to explore the capability of T. militaris to adapt to ongoing environmental change in order to maintain extant populations.

In this study we investigated patterns of population genetic structure in T. militaris using genotype-by-sequencing and a sampling regime encompassing the species’ entire distribution spanning seven degrees of latitude and a difference of 3.9 °C in mean annual sea surface temperature. We explore patterns of gene flow and population connectivity across the sampling distribution and integrate genomic and geospatial data to tests for genotype-environmental associations (GEAs) indicative of adaptive genetic differentiation between populations. Findings from this study provide valuable insights into the spatial scales of gene flow and the availability of standing genetic variation for adaptation to future environmental conditions. We discuss the findings of this study in the context of future fisheries management.

Materials and Methods

Sampling

Eight rocky shore locations (0 to 5 m depth) were selected for sampling (Table 1) spanning the known range of T. militaris from Hastings Point (northern NSW) to Jervis Bay (southern NSW) (Fig. 1), representing a seven-degree latitudinal and a 3.9 °C annual mean sea surface temperature gradient. Additionally, to sample a known cross-shelf gradient of temperature (less confounded by distance and unconfounded by latitude), three locations at varying distances from shore and under different temperature regimes were selected: Nambucca Heads (mainland), Split Solitary Island (3 km from the mainland) and South Solitary Island (7 km from the mainland). These represent inshore, mid-shelf, and offshore positions, respectively, with an annual mean sea temperature varying by about 1 °C from inshore to offshore (Malcolm, Jordan & Smith, 2010). Notably, South and Split Solitary islands are remote from shore and are also likely to have very low to no harvesting pressure. Between 25 and 30 T. militaris individuals were collected from each location (Table 1). As internal foot muscle tissue was found to yield clean, high molecular-weight DNA, it was necessary to narcotise specimens. Field collections were carried out under a scientific collection permit granted by the NSW DPI Fisheries under section 37 of the NSW Fisheries Management Act 1994 (#P01/0059(A)-4.0).

Table 1 Details of samples, sampling location and location based environmental covariables for Turbo militaris in New South Wales.

Location	Latitude	Longitude	Date collected	N samples	SST (°C)	Temp range (°C)	EKE	CHL a	
Hastings Point	28°21′38.09″S	153°34′44.21″E	30 Mar 2021	25	23.35	8.91	0.1391	2.1094	
Woody Head	29°21′47.70″S	153°22′25.96″E	08 Apr 2021	25	22.34	9.52	0.0632	3.8472	
South Solitary Island	30°12′19.31″S	153°16′01.24″E	12 May 2021	25	22.42	10.17	0.1204	0.9517	
Split Solitary Island	30°14′24.07″S	153°10′50.71″E	12 May 2021	30	22.37	10.14	0.0741	2.0323	
Nambucca Heads	30°38′47.49″S	153°01′15.10″E	26 Mar 2021	23	22.23	10.23	0.1223	2.0319	
Crowdy Head	31°50′16.91″S	152°45′04.54″E	28 Apr 2021	22	21.34	9.94	0.0820	2.7083	
Newcastle	32°57′31.70″S	151°45′02.72″E	12 Apr 2021	31	20.39	9.12	0.0178	2.2840	
Jervis Bay (Plantation Pt)	35°06′16.15″S	150°41′53.70″E	20 Aug 2021	25	19.46	10.24	0.0311	2.9248	
Total				206					
Note:

N, number; SST, sea surface temperature; Temp range, temperature range; EKE, eddy kinetic energy; CHLa, chlorophyll a.

Figure 1 Map of Turbo militaris collection sites from the eastern seaboard of Australia.

DNA extraction and genotyping

Approximately 25 mg of foot muscle tissue was sampled from each specimen using sterilised scalpel and forceps, avoiding the inclusion of mucous-rich epidermal tissue. Tissue samples were immediately placed in 2.0 mL Eppendorf snap-lock microcentrifuge tubes containing 500 µL of hexadecyltrimethylammonium bromide (CTAB) lysis buffer (100 mM TrisHCl, 20 mM EDTA, CTAB 2% w/v, NaCl 1.5 M) and refrigerated at 4 °C for 2 weeks. Proteinase K (30 µL at 20 mg/mL) was added to the samples which were incubated in a Allsheng shaking incubator at 60 °C, 200 rpm overnight. Samples were cooled to room temperature and purified by addition of an equal volume of 24:1 chloroform isoamyl. After centrifugation, the aqueous phase was retained and DNA was precipitated by the addition of 800 µL of dilute CTAB buffer (100 mM TrisHCl, 20mM EDTA, CTAB 2% w/v) to each tube which were incubated in an Aosheng MSC-100 shaking incubator at 60 °C at 400 rpm until DNA/CTAB complexes were visible. After centrifugation, the DNA formed a pellet which was twice cleaned with 70% ethanol by repetitive inversion. The DNA pellet was air-dried to remove residual ethanol and subsequently resuspended in sterile lab-grade water. Extracted DNA was quality checked using NanoDrop, Qubit assay and gel electrophoresis.

For single nucleotide polymorphism (SNP) genotyping, 20 μL of extracted DNA was sent to Diversity Arrays Technology Pty Ltd (Canberra, Australia) (DArT). The DArT organisation provides a process pipeline of whole-genome profiling, without the need for a reference genome (Jaccoud et al., 2001). High-throughput DArTseq technology was used to genotype Turbo militaris DNA. Here, the PstI-based complexity reduction method (Wenzl et al., 2004) was applied for the enrichment of genomic representation with single copy sequences. This method involved the digestion of DNA samples with a cutting enzyme PstI, paired with a set of secondary frequently cutting restriction endonucleases, ligation with site-specific adapters, and amplification of adapter-ligated fragments. Post digestion with a restriction enzyme pair, a PstI-overhang-compatible oligonucleotide adapter was ligated, and the adapter-ligated fragments were amplified in adherence to standard protocol (Wenzl et al., 2004). To develop SNPs, the DArTseq technology was optimized using two PstI-compatible adapters corresponding to two different restriction enzyme overhangs. The genomic representations were generated following the procedures described by Kilian et al. (2012). Next-generation sequencing technology was implemented using HiSeq2000 (Illumina, USA) to detect SNP markers. Sequence data was analysed using DarTsoft14 and DArTdb (Kilian et al., 2012).

SNP calling

In total, 208 individual T. militaris were initially genotyped with the DArTseq™ platform yielding a total of 19,837 SNP loci with a mean read depth of 15.31 and 10.73% missing data. To improve SNP quality, while optimising the number of loci available for population genomic analyses, quality control filters, based on the descriptive statistics from the DArTseq™ pipeline, were applied to data using the R package dartR v.2.7.2 (Gruber et al., 2019; Mijangos et al., 2022). Prior to SNP quality control, we checked for the presence of genetically related individuals, as their inclusion can lead to biased genetic estimations of downstream analyses. We calculated a similarity genetic distance matrix for individuals on the proportion of shared alleles per pairs of individuals with the function gl.propShared in dartR. Two individuals, one from Nambucca Heads (NAM07) and one from South Solitary Island (SSI16) were characterised as closely related and were removed from downstream analyses.

Two SNP datasets were generated for analyses: dataset ‘1’ for the analysis of overall genetic structure, and dataset ‘2’ for tests of GEAs and the identification of candidate loci. Both datasets were generated by retaining a single SNP per tag, removing secondaries, applying a locus and individual call rate of 80%, reproducibility threshold of 80%, and a Hamming distance threshold of 0.2 to control for the influence of linkage disequilibrium between loci. SNPs were called for dataset ‘1’ by applying a minor allele frequency (MAF) threshold of 0.03 and removing all loci departing from Hardy–Weinberg expectations. In contrast, SNPs were called for dataset ‘2’ by setting MAF to 0.01, and not filtering out SNP loci deviating from by Hardy–Weinberg expectations (all loci included). Finally, we also used poppr (Kamvar, Tabima & Grunwald, 2014) in the R package to calculate the number of private alleles found in each population, and remove these SNPs using gl.drop.loc in package dartR. After filtering, a total of 3,527 and 6,852 SNP loci for 206 individuals (Table 2) were retained for data sets ‘1’ and ‘2’, respectively.

Table 2 Predictor variables sourced from copernicus marine environment monitoring service used in genotype by environmental association analyses.

Predictor	Description	Spatial resolution	Units	
SST	Daily global sea surface temperature reprocessed (level 4) from Operational SST and Ice Analysis system downloaded from CMEMS (product #010_011).	0.05°	°C	
CHLa	8-day composite mass concentration of chlorophyll a in seawater (level 4) from Globcolour downloaded from CMEMS (product #009_082).	4 km	mg m−2	
EKE	Daily eddy kinetic energy computed from zonal and meridional velocity components from the Sea Level Thematic Assembly Centre downloaded from CMEMS (product #008_047).	0.25°	m2s−2	

Tests for population differentiation

Several estimates of genetic diversity were generated using the poppr package in R, including observed (HO) and expected (HE) heterozygosity and allelic richness (AR) (Joop Ouborg, Angeloni & Vergeer, 2010). For AR, allele counts were rarefied by the minimum number of individuals genotyped using the allelic.richness command in the R package hierfstat (Weir & Goudet, 2017). Statistical differences in genetic diversity measures among sites was estimated using the Hs.test function in the R package adegenet (Jombart, 2008). Departures from random mating were calculated using FIS (inbreeding coefficient) for the overall dataset and for each sample location using the basic.stats function in hierfstat.

Tests for population genetic structure were subsequently performed, by calculating global population differentiation (FST) with 95% confidence limits (Weir & Cockerham, 1984), and population pairwise measures of FST with significance determined using permutation (999) in the dartR. Multiple testing effects were corrected for using the Benjamini–Hochberg FDR procedure (Benjamini & Hochberg, 1995). An analysis of molecular variation (AMOVA) was performed in the R package poppr (Kamvar, Tabima & Grunwald, 2014), using a model that partitioned variation among sample sites and within sample sites, with significance based on a randomization test with 999 permutations. A mantel test of isolation by distance (IBD) was carried out using the gl.ibd (Rousset, 1997) function in the R package dartR (Gruber et al., 2019) with results visualised in a scatterplot. A discriminant analysis of principal components (DAPC) was performed using adegenet package in R (Jombart, 2008). The find.clusters function was used to detect the number of clusters in the population. The best number of subpopulations has the lowest associated Bayesian Information Criterion (BIC). A cross validation function (Xval.dapc) was used to confirm the correct number of PCs to be retained. Finally, we used sparse non-negative matrix factorisation (sNMF) implemented in the R package LEA (Frichot & François, 2015) in R. This algorithm estimates the genetic ancestry components for each sample. For this study, 15 runs were performed with α = 100 for each K value (one to eight). The selection of the best number of putative ancestral populations was guided by the cross-entropy criterion (wherein, for K, a plot of the cross-entropy curve formed a ‘knee’) and the results from the best run were visualised using the barplots function.

Environmental variables

Physical oceanographic data were downloaded from the Copernicus Marine Environment Monitoring Service (https://marine.copernicus.eu), using a 20-year historical time-series encompassing 2001–2020 (daily temporal resolution) matched to each sampling location. These variables included temperature, water flow and productivity: sea surface temperature (SST) at 0.05° spatial resolution; remotely-sensed eddy kinetic energy (EKE) at 0.25° spatial resolution; and remotely-sensed chlorophyll a concentration (CHLa) at 0.04° spatial resolution (Table 2). The native spatial resolutions of oceanographic variables were used when matching daily data to sampling locations. Mean values were calculated for each variable by averaging the daily data. In addition, the lowest and highest SST for each location was extracted, and an absolute temperature range variable (temp.range) was calculated by subtracting the minimum from the maximum. A pairs.panels scatter plot of matrices was generated in the R package psych to confirm a lack of (Pearson’s) correlation between variables (r2 ≤ 0.8), which were subsequently used in GEA analyses (described below).

Genome–environment association (GEA) analyses

To detect putative genomic signatures of selection, tests for GEAs were conducted using two complementary models: latent factor mixed models (LFMM2) (Caye et al., 2019), and the Bayesian method available in BayPass V1.01 (Gautier, 2015). BayPass and LFMM accounted for potentially confounding allele frequency differences due to population structure in a mixed linear model framework, but in different ways: LFMM estimates GEAs when simultaneously correcting for population structure with latent factors, while BayPass uses a neutral covariance matrix constructed from population allele frequencies. The subset of environmental variables described above were used as predictor variables for both analyses.

Genotype-environment associations were explored with Baypass V1.01 (Gautier, 2015) under the auxiliary (AUX) covariate model (-covmcmc and -auxmode flags). The first core model (without the environmental data) was run to estimate a covariance matrix (Ω) of population allele frequencies, which is an approximation of genomic differentiation between populations caused by demographic history. In order to reach convergence and reproducibility of the MCMC estimates, five independent runs, each with a randomly chosen seed were performed using default parameters, except for: pilot runs length of 1,000 iterations, number of sampled parameter values of 1,000, and a burn-in period length of 2,500 iterations. Secondly, the average of the five covariance matrices were used as input for the auxiliary covariate model to detect evidence of an association corrected for population structure. Environmental variables were scaled using the “-scalecov” option and the same running parameters as the core model were applied. Finally, the strength of association between genotype and the covariates was assessed by calculating the average of the log-transformed Bayes Factor (BF) in deciban units (dB) for each locus and environmental predictor. Significance was determined following Jeffrey’s criterion for decisive associations (BFis ≥ 20) (Jeffreys, 1939).

Latent factor mixed models (LFMM2) tested for linear relationships between environmental variables and genetic variants with random latent factors using a least-square method. Population structure was inferred by estimating individual ancestry coefficients based on sparse non-negative matrix factorisation (SNMF) method implemented in the snmf function in the R package LEA v3.10.2 (Frichot & François, 2015). Ancestry coefficients were determined for 1–8 ancestral populations (K) by generating an entropy criterion that evaluates the fit of the statistical model to the data using a cross-validation technique (Frichot & François, 2015). The K with the lowest cross-entropy value using 100 repetitions for each K value was selected. Subsequently, the optimal factor, K = 1, was used to inform the LFMM to identify whether allele frequencies were correlated with any of the environmental variables. To increase the statistical power of associations, missing genotype data were imputed via the ‘impute’ function in the LEA package, using the most common allele frequency observed in each K with the method ‘mode’. Next, we used the function lfmm_ridge to compute a regularised least-squares estimate using a ridge penalty. Individual associations between each SNP frequency and each environmental variable were assessed using statistics test calibrated using genomic inflation factor (function lfmm_test). Corrections for multiple comparisons were applied with the Benjamini-Hochberg algorithm with a false discovery rate (FDR) threshold of 5% (Benjamini & Hochberg, 1995). Significance associations were determined using a threshold of 0.001, since the probability of finding a false positive result increases with lower thresholds (Ahrens, Byrne & Rymer, 2019).

The gradient forest (GF) algorithm was subsequently used to describe the strength of the associations of spatial, environmental variables and candidate loci and to map spatial patterns of allelic turnover in climate space (Ellis, Smith & Pitcher, 2012; Fitzpatrick & Keller, 2015). Gradient forest is a machine learning method initially developed to model the turnover of ecological community assemblages in relation to environmental gradients (Ellis, Smith & Pitcher, 2012). Recently, this method has been adapted as a landscape genomics toolbox, substituting allele frequencies at genetic loci for species to model allelic turnover in climatic space (Fitzpatrick & Keller, 2015). The turnover functions in gradient forest allow for inference of the environmental predictors driving observed changes in allele frequency (Fitzpatrick & Keller, 2015). Analyses were implemented in the R package gradientForest (Ellis, Smith & Pitcher, 2012), using a regression tree-based approach to fit a model of responses between genomic data and environmental variables (Capblancq et al., 2020). Specifically, adaptive genetic variation turnover were modelled on the seascape using the candidate SNPs (derived from LFMM, and BayPass) set as the response variables. The machine learning algorithm partitioned allele frequencies at numerous splits values along each environmental gradient and calculated the change in allele frequencies for each split. The split importance (i.e., the amount of genomic variation explained by each split value) was cumulatively summed along the environmental gradient and aggregated across alleles to build a non-linear turnover function to identify loci that are significantly influenced by the predictor variable (Ellis, Smith & Pitcher, 2012). The analysis was run over 500 regression trees for each of the four environmental variables with all other parameters at default settings. The cumulative goodness-of-fit among SNPs was represented as an R2 value indicating how well a predictor explained changes in allele frequency and which predictors were most important in predicting genomic changes. The resulting multidimensional genomic patterns were summarised using principal component analysis (PCA), allowing the relative importance of predictor variables on allelic turnover to be visualised. Finally, using the top gradient forest model, we interpolated genetic composition and allelic turnover across the sampling range of eastern Australia.

Results

Overall population genetic structure

Patterns of genetic diversity did not differ greatly across the eight sample locations (HO = 0.186–0.212; AR = 1.630–1.640; Table 3). Most sites showed a weak excess of heterozygotes (FIS = 0.048–0.118), however these estimates did not differ significantly from zero (p > 0.01; Table 3). Overall genetic differentiation was found to be significant, but weak and close to zero (global FST = 0.002, p < 0.001) indicating a lack of genetic structure among sampling locations (Table 3). These findings are further supported by weak, yet significant, estimates of genetic differentiation among all pairs of sampling locations (FST = 0.000–0.004, Table 4). AMOVA also indicated a lack of overall genetic structure indicating genetic variance attributed to differences among sites to be non-significant (0.212, p > 0.05) while the majority of variance was explained by genetic variation between individuals within sites (99.86, p > 0.05). Similarly, DAPC and LEA analyses indicated a lack of genetic structure, both identifying a single population cluster (K = 1). Finally, Mantel tests revealed no significant relationship between genetic differentiation and distance between sampled locations (R2 = 0.03356, p = 0.278) providing further evidence of panmixia.

Table 3 Summary of descriptive statistics.

Location	n	A R	H E	H O	F IS	
Hastings Point	25	1.635	0.218	0.190	0.110	
Woody Head	25	1.633	0.216	0.189	0.107	
Split Solitary Island	30	1.632	0.217	0.189	0.111	
South Solitary Island	25	1.638	0.219	0.202	0.076	
Nambucca Heads	23	1.640	0.221	0.212	0.048	
Crowdy Head	22	1.633	0.217	0.188	0.111	
Newcastle	31	1.630	0.215	0.186	0.118	
Jervis Bay	25	1.630	0.217	0.194	0.096	
Note:

Sample size (n), mean allelic richness (AR), and genetic diversity indices including expected (HE) and observed (HO) heterozygosity, and inbreeding coefficients (FIS) at each site, based on the complete filtered dataset (n = 207 individuals).

Table 4 Pairwise estimates of pairwise FST among sample locations.

	Nambucca Heads	Woody Head	Newcastle	Hastings Point	South Solitary Is	Crowdy Head	Split Solitary Is	Jervis Bay	
Nambucca Heads		0.0037	0.0037	0.0031	0.0033	0.0028	0.0028	0.0043	
Woody Head	0.00		0.0022	0.0019	0.0034	0.0029	0.0027	0.0038	
Newcastle	0.00	0.00		0.0017	0.0024	0.0016	0.0014	0.0021	
Hastings Point	0.00	0.00	0.01		0.0018	0.0007	0.0014	0.0017	
South Solitary Is	0.00	0.00	0.00	0.00		0.0016	0.0020	0.0031	
Crowdy Head	0.00	0.00	0.00	0.11	0.00		−0.0002	0.0016	
Split Solitary Is	0.00	0.00	0.00	0.00	0.00	0.69		0.0020	
Jervis Bay	0.00	0.00	0.00	0.01	0.00	0.00	0.00		
Note:

Bolded FST values were found to be significant after multiple corrections (p < 0.05).

Genotype x environment associations

BayPass and LFMM each identified a number of candidate SNP loci exhibiting significant genotype-by-environment associations for each of the environmental predictors tested (Fig. 2). BayPass detected between 0 and 2 SNPs with significant correlations (log10(BF) > 20) for each of the environmental predictors, with zero overlap in candidates between predictors. In contrast, LFMM detected between 5 and 12 SNPs that were significantly correlated with each of the environmental predictors (Fig. 2), but with only two loci overlapping between the EKE and SST predictor variables. Concordance of candidate SNP loci between methods was low (three loci only; Fig. 2) but expected given these methods have varying sensitivities to detecting loci under selection, use different methods for controlling for demography, and adopt different association algorithms. Given that T. militaris exhibits panmixia, its lack of population structure was likely to influence the inference of omega matrix and K clusters on BayPass and LFMM respectively. Furthermore, LFMM tests for relationships between individual-based allele frequencies whereas BayPass is at population level. Additionally, sampling design is likely to have had an influence, wherein a strategic sampling design accommodated to environmental heterogeneity and spatial variation on a landscape is essential to potentially identify and validate patterns of local adaptations across natural populations. Overall, GEAs indicated that a larger proportion of SNPs were significantly associated with temp range (13) and CHLa (11), followed by EKE (7) and SST (5).

Figure 2 Results of genotype x environment association analyses show the strength of associations between individual loci and environmental variables identified by BayPass and LFMM2.

Bar plots show the number of putative candidate loci associated with each environmental variable.

Gradient Forest modelling used the unique candidate adaptive loci detected by both LFMM and BayPass (Luo et al., 2021), with 39 in total, four in BayPass and 35 in LFMM, but three were shared between both analyses and one was shared within LFMM (variables EKE and SST) (thus 39—3—1 = 35 unique candidate loci). Gradient forest analyses showed 14 of the 35 candidate SNP loci to be significantly correlated with environment (R2 values > 0; mean = 0.08, range 0.003–0.3). Overall, EKE, CHLa and SST were found to be the most important predictors of genomic variation, while temp range had less of an effect (Fig. 3). Turnover functions from the GF model show the weighted cumulative importance values and sharp turnovers for all environmental predictors, but again with temp.range having lower importance relative to all other predictors (Fig. 3A). Biplots based on the first two principal components captured approximately 99% of the total variation and point to EKE and SST as the most prominent drivers of genomic variation (Fig. 3B). A spatial depiction of genomic composition in multi-dimensional climatic space based on PCA is provided in the allelic turnover map (Fig. 3C). The map indicates that the turnover of putatively adaptive allelic variation tracks closely with latitude, with the genomic composition of northern and southern most sampling locations being distinct from those from geographically intermediate locations (Fig. 3D).

Figure 3 Gradient Forest outputs.

(A) Overall relative importance for each environmental variable describing the turnover in allele frequencies from the Gradient Forest model; (B) cumulative importance curves showing overall pattern of genomic compositional change (R2, y-axis) for each environmental gradient (x-axis). Turnover functions for each curve are aggregated across all candidate loci. The curve shape indicates the rate of change in allele frequencies along the environmental gradient, and the maximum height indicates the total turnover in allele frequencies. Predictor variables shown here include: chlorophyll a concentration (CHLa), eddy kinetic energy (EKE), sea surface temperature (SST) and sea surface temperature range (temp.range); (C) Influence of environmental variation on candidate SNP allele frequencies inferred from Gradient Forest analyses as aPCA plot illustrating the influence of environmental variation on allele frequencies for candidate loci. Background colours denotes environmental space, whereas black dots represent the principal component (PC) scores associated with the sampling locations; (D) Predicted spatial turnover in allele frequencies interpolated across the study area, with black dots as labelled sampling locations. Similar colours indicated areas expected to have similar genetic composition, while divergent colours indicate divergent putatively adaptive genotypes. Colours are based on the first three principal components of transformed environmental variables.

Discussion

Understanding spatial patterns of gene flow and local adaptation can help predict species responses to climate change and to identify populations most at risk of maladaptation (Hoffmann & Sgrò, 2011; Sexton et al., 2009). Such information is critically important for assisting with the adaptive management of commercially important marine species, many of which are already showing signs of climate stress (Cheung et al., 2013; Pinsky et al., 2018; Sunday et al., 2015). This study represents the first population genomic analysis of T. militaris, a commercially and culturally important trochid marine gastropod from the east coast of Australia, with the purpose of informing fisheries managers about vulnerability of this species to future climate change. Analyses of SNP genotypes across the species’ entire distribution spanning seven degrees of latitude and 3.9 °C in mean annual sea surface temperature indicate the presence of a single admixed, and potentially panmictic, demographic unit with no evidence of genetic subdivision along the entirety of its range. Furthermore, significant genotype associations with heterogeneous habitat features were observed at regional spatial scales, including associations with sea surface temperature, ocean currents, and nutrients, indicating possible adaptive genetic differentiation among sample locations. Combined, these findings provide insights into the potential resilience of T. militaris to changing marine climates and the potential influence of gene flow and selection on future adaptive responses.

Evidence of panmixia

Our analyses point to a lack of genetic structure across the entire distribution of T. militaris indicating widespread gene flow along the eastern seaboard of Australia. Such genetic patterns are also found amongst other trochoid taxa with long pelagic larval phases which are expected to facilitate long distance dispersal (Berry et al., 2019; Díaz-Ferguson et al., 2010; Nikula, Spencer & Waters, 2011; Silliman, Grosholz & Bertness, 2009). Several other eastern Australian marine invertebrates also exhibit high gene flow facilitated by larval traits including the Crown-of-thorns sea star (Acanthaster spp.) (Pratchett et al., 2015), the surf bivalve Donax deltoides Lamarck, 1818 (Murray-Jones & Ayre, 1997; Miller et al., 2013) and the black sea-cucumber Holothuria (Mertensiothuria) leucospilota (Brandt, 1835) (Chieu et al., 2023). Here gene flow and population structure has been linked to the long-distance dispersal of pelagic larvae facilitated by a fast-flowing East Australian current (EAC). While knowledge of the reproductive biology or larval competency of T. militaris is poor, our results suggest that the species may also generate long-lived, planktotrophic larvae contributing to high levels of biological connectivity across its distribution (Cowen & Sponaugle, 2009). Overall, these findings are consistent with previous genetic studies on trochoid taxa indicating population admixture over vast geographical areas. In this case we have provided evidence of gene flow among potentially locally adapted populations spanning major environmental gradients.

Evidence of local adaptation

Despite a lack of overall genetic structure across the sampling distribution, significant genotype associations with heterogeneous habitat features were observed across the sampling distribution, including associations with annual mean sea surface temperature, sea surface temperature range, EKE and productivity (CHLa). Drift processes leading to neutral genetic structure are often suppressed in broadcast spawning marine organisms with large population sizes (Gélin et al., 2017; Palumbi, 2003; Pinsky & Palumbi, 2014), but numerous studies have shown that adaptive genetic divergences can still be established and maintained under strong selection pressure (Hendry, 2017; Nosil, 2012; Schluter, 2000). In fact, many studies have demonstrated that adaptive variation can be maintained despite high levels of gene flow specifically in marine invertebrates, including gastropod snails (Miller et al., 2019; Sandoval‐Castillo et al., 2018). However, these findings are based on correlative tests only and should be interpreted with caution, as controlled mechanistic experiments are needed to validate these patterns and drivers of putative adaptive differentiation (Savolainen, Lascoux & Merilä, 2013; Stinchcombe & Hoekstra, 2008). Also, while our sampling regime was designed to correct for geographical distance, we cannot rule out the possible influence of artefactual associations (i.e., false positives) and SNP associations with other environmental factors varying with latitude (such as light intensity, dissolved oxygen levels and rainfall-driven variation in salinity). Nevertheless, our findings are consistent with many studies of marine species which demonstrated genetically determined clines related to climatic variables in Australia and overseas (Poloczanska et al., 2013, 2016; Wernberg et al., 2016).

Resilience to future climatic challenges

The coastal waters of south-eastern Australia are a recognised climate change hotspot with warming occurring above the global average with changes enhanced by strengthening of the EAC (Cresswell, Peterson & Pender, 2016; Malcolm et al., 2011; Oliver et al., 2017). Predictions anticipate warming to continue to increase, generating deteriorating marine conditions for sessile taxa that are already at their thermal limits (e.g., Davis, Champion & Coleman, 2021). Within the EAC, the effect of warming and changes to circulation may also counter one another, with warming enhancing larval survival, but a strengthened current reducing larval supply to the coast by restricting cross-current larval dispersal (e.g., in lobster, Cetina-Heredia et al., 2015). Our findings suggest that T. militaris may have the capacity to adapt to future climatic challenges, assisted by both widespread gene flow across environmental gradients and the availability of standing genomic variation for selection to act upon. Findings of putative adaptive genetic differentiation associated with temperature, nutrients and ocean currents is a particularly important finding, suggesting that standing genetic variation may be available for selection to act on to counter future environmental change, assisted by widespread gene flow, high fecundity (Romolo & Trijoko, 2021) and highly probable short generation time in this species (Kimani, 1996; Romolo & Trijoko, 2021).

While our findings suggest that T. militaris is likely to be generally resilient to shifts in the physical ocean climate, aggregations in some areas may still be vulnerable to risks of maladaptation. Those most at risk are likely to include locally adapted populations, where projected local changes in climate are high, and connections to non-local aggregations are relatively weak (Hoffmann & Sgrò, 2011). Recent climate projections indicate that many low energy embayment habitats are likely to experience greater increases in SST than open coastal habitats (Guyondet et al., 2015; Scanes, Scanes & Ross, 2020; Vila-Concejo et al., 2007); Also, biophysical models suggest that the biological connections between low energy embayment and high energy open coastal habitats can be weak in some marine invertebrates from south-eastern Australia (Riginos et al., 2016; Treml et al., 2015). Consequently, it is possible that locally adapted aggregations from low energy embayment habitats may be most vulnerable to climate change effects, where gene flow is unlikely to assist local aggregations in adapting to warming sea surface temperatures via the migration of thermally adapted genotypes. In such cases adaptive management strategies might be needed, including the assisted migration of thermally adapted genotypes to populations showing signs of climate stress. Such approaches are being widely advocated as a tool for “climate proofing” threatened marine and terrestrial animal and plant communities (Aitken & Whitlock, 2013; Prober et al., 2015; Layton et al., 2020; Hoffmann, Miller & Weeks, 2021). Although the northernmost Hastings Point population is not genetically isolated, its position at the northern trailing edge may render it vulnerable to stochastic events, such as heatwaves (Ab Lah et al., 2018; Mamo et al., 2019). Vulnerability in trailing edge populations can be amplified by genetic impoverishment through loss of individuals, without replenishment through immigration (gene-flow), consequently exposing these populations to the risk of localised extinction (Clark et al., 2020; Coleman et al., 2020). Indeed, this species has undergone prior range shifts indicating that the trailing (warm) edge is likely to be vulnerable to ongoing warming.

Implications for fisheries management

Sustainable fisheries management requires information on factors that influence the resilience of individual fishing stocks to fishing pressure and environmental disturbance (Astles et al., 2006; Kenny et al., 2018). This includes understanding the geographic boundaries of biological populations and the recruitment potential of individual stocks persisting within and across these populations (Binks et al., 2019; Roughgarden, Iwasa & Baxter, 1985) and how this might be altered under climate change. In the case of T. militaris, the presence of a single panmictic population unit across its distributional range, generally means that the opportunity for recolonisation following depletion events (overharvesting or environmental disturbance) are enhanced for central and southern sub-populations. However, for the northern trailing edge, the opportunity for repopulation with genotypes from pools of genetic diversity further south may be hampered by the dominant poleward flow of the EAC. Furthermore, increasing human population and harvesting pressure may reduce local abundance (Cooling & Smith, 2015) and the opportunity for thermal adaption and repopulation.

Conclusions

Knowledge of population genomics, particularly adaptive structure, is important for fisheries management and can be used to estimate vulnerability and adaptability of stocks under climate change. This study revealed that the harvested gastropod, T. militaris, is panmictic across its distributional range with little variation in genetic diversity and can be considered as a single stock. As such, it has the genetic capacity to survive and proliferate within its environmental niche and is likely to continue to track ocean temperatures by shifting its entire distribution poleward. Genomic studies can improve management of harvested species under climate change by providing insights into adaptive capacity and help identify opportunities for strategic adaptive management (van Oppen & Coleman, 2022).

Supplemental Information

Supplemental Information 1 Turbo militaris R Script Filtering for Population genetics.

Click here for additional data file.

Supplemental Information 2 Turbo militaris R Script Filtering for GEA.

Click here for additional data file.

Supplemental Information 3 Turbo militaris R Script Run BayPass.

Click here for additional data file.

Supplemental Information 4 Raw SNP dataset -output for DArT Next Gen genotyping.

Click here for additional data file.

Supplemental Information 5 Turbo militaris R Script Run LFMM2.

Click here for additional data file.

Supplemental Information 6 Curated environmental data Turbo militaris analysis.

Click here for additional data file.

Supplemental Information 7 Turbo militaris R Script Run GF.

Click here for additional data file.

We acknowledge the assistance of Matthew Hammond for field collection at Jervis Bay, NSW.

Additional Information and Declarations

Competing Interests

Author Contributions

Field Study Permissions

Data Availability

The authors declare that they have no competing interests. Stephen D. A. Smith is employed by Aquamarine Australia.

Matt J. Nimbs conceived and designed the experiments, performed the experiments, analyzed the data, prepared figures and/or tables, authored or reviewed drafts of the article, fieldwork, and approved the final draft.

Curtis Champion analyzed the data, authored or reviewed drafts of the article, data acquisition, and approved the final draft.

Simon E. Lobos analyzed the data, prepared figures and/or tables, authored or reviewed drafts of the article, and approved the final draft.

Hamish A. Malcolm conceived and designed the experiments, authored or reviewed drafts of the article, fieldwork, and approved the final draft.

Adam D. Miller conceived and designed the experiments, performed the experiments, analyzed the data, prepared figures and/or tables, authored or reviewed drafts of the article, and approved the final draft.

Kate Seinor conceived and designed the experiments, authored or reviewed drafts of the article, fieldwork, and approved the final draft.

Stephen D. A. Smith analyzed the data, authored or reviewed drafts of the article, fieldwork, and approved the final draft.

Nathan Knott conceived and designed the experiments, authored or reviewed drafts of the article, fieldwork, and approved the final draft.

David Wheeler performed the experiments, analyzed the data, authored or reviewed drafts of the article, and approved the final draft.

Melinda A. Coleman conceived and designed the experiments, performed the experiments, authored or reviewed drafts of the article, fieldwork, and approved the final draft.

The following information was supplied relating to field study approvals (i.e., approving body and any reference numbers):

Field collections were carried out under a scientific collection permit granted by the NSW DPI Fisheries under section 37 of the NSW Fisheries Management Act 1994

The following information was supplied regarding data availability:

Raw SNP data used for analytical purposes is available at Dryad: Nimbs, Matt J. (2023). Single Nucleotide Polymorph and location metadata for Turbo militaris from Eastern Australia [Dataset]. Dryad. https://doi.org/10.5061/dryad.79cnp5j27.

The data processing R scripts are available in the Supplemental Files.

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
