# Peer review of "Genomic analyses indicate resilience of a commercially and culturally important marine gastropod snail to climate change"

_PeerJ, doi:10.7717/peerj.16498_

## Round 0.1 · original submission · Major Revisions

While the reviewers were overall positive about the manuscript, they have identified a range of revisions that are required, especially the lack of data availability requires addressing.

Reviewer 1 ·

Basic reporting

• The manuscript is well written in professional and clear English. There are a few minor spelling mistakes present that should be corrected.
o L66: change “environmental” to “environments”
o L78: delete “the” before “deliberately”
o L110: full stop missing
o L112: “being targeted”
o L121: “Information” should be lower case
o L133: Maybe add a delta sign in front of 3.9C to indicate that this is the differ-ence in sea surface temperature? Also, there is a discrepancy in the eight degree latitudinal gradient mentioned here and the seven degree latitudinal gradient men-tioned in the methods. Which statement is correct?
o L138: Replace “content” with “context”
o L188: “calculated”
o L191: “were removed from downstream analyses”
o L250: “account”
o L272: “test”
o L290: “Significant associations were”
o L328: “pairs of”
• The background is well structured and provides a good introduction into the topic. Ref-erences are appropriately cited.
• Field study permits were obtained.
• Overall, the figures represent the main results well, but might need some more infor-mation in the captions. E.g., what do the colors and the inset in Figure 2 represent?
• I might have missed this, but I did not see a data availability statement in the manuscript. Please provide all raw sequence data and bioinformatic/analysis scripts used to make this study reproducible.

Experimental design

• Molecular and statistical methods are appropriately chosen, though in some cases it might be good to explain their goal a bit more for the unfamiliar reader. E.g., L213-232 has a list of analyses but it is not explained why they are performed or how they com-plement each other. Also, for the GEA analyses it does not become immediately clear what additional information the gradient forest analysis brings. I would suggest adding an introductory sentence on how it is augmenting the BayPass and LFMM analyses.
• L287: Could you mention the statistical test used here specifically?
• L337ff: Please mention here the total number of SNPs that were detected by each meth-od. Overall, I think this section needs more explanation (and justification) of how the fi-nal set of candidate SNPs was chosen given that the two methods agreed relatively poorly. From the number of SNPs that were inputted into the gradient forest analysis it looks like the SNPs from the LFMM analyses were used (35 in total)? Could you ex-plain why you made this choice and why you think these SNPs are not false positives? Optimally you would reference a paper here that can support higher accuracy of the LFMM method over BayPass.

Validity of the findings

• While I think the GEA analyses are very interesting, I am afraid they are overinterpreted. The amount of variance explained by each tested predictor in the gradient forest analysis is extremely small, the candidate SNPs do not agree well between methods and are in to-tal very few. Similarly, the importance of predictors does not agree between the LFMM/BayPass and gradient forest analyses. For example, temperature range seems to have the highest number of associated SNPs but is the weakest predictor in the gradient forest analysis (possibly indicating that some candidate SNPs are false positives). Over-all, I think evidence for adaptation is weak in this dataset, in agreement with the absence of genetic structure. I would advise the authors to tone this result down significantly and restructure the discussion and conclusions accordingly. To me it seems that the absence of genetic structure over the investigated environmental gradient indicates phenotypic plasticity rather than local adaptation and that this plasticity would likely foster resilience of the species to climate change in addition to the strong presence of gene flow and connectivity (maintaining genetic diversity) within the species’ distributional range.

Reviewer 2 ·

Basic reporting

The manuscript is largely well written with logical flow and sufficient background and context. The authors also cite a plethora of relevant literature and present the article in professional manner. One criticism I have with respect to the background, is that the authors repeatedly state that gene flow is beneficial to local adaptation but we know this isn't always the case unless selection is strong enough to maintain these adaptive alleles. The authors touch on this in the Discussion but it should also be mentioned in the Introduction. Additionally, it seems that throughout the manuscript the authors invoke panmixia as evidence for climate change resilience/adaptive capacity, and more specifically, the role of standing variation in facilitating this. I don't think you can make this inference or conclusion unless you look at the number of private alleles and other metrics of genetic diversity (i.e. nucleotide diversity). At present, your values of heterozygosity are very low (and observed is consistently lower than expected) suggesting that there isn't a lot of genetic diversity within the system to buffer against environmental change. There are also issues with data accessibility- the authors haven't reported where the genomic and environmental data are stored. Below, I detail additional revisionary suggestions:
Line 60: 'at' instead of 'an'
Lines 63-70: I think the writing could be improved here- the message is lost. For instance, it's not clear what you're referring to when you say 'This typically applies to less vagile organisms...' and '...standing genetic variation in quantitative traits...'
Line 78: 'deliberate' instead of 'deliberately'
Lines 82-84: Suggested revision: "Population genetics has played a key role in characterising structure and gene flow among populations, especially in commercially important marine species."
Line 92: 'Fish stocks' instead of 'fishing stocks'
Line 118: Presumably these other studies used much smaller datasets (a few microsats, perhaps) so it would be worth mentioning the need to scale up efforts and survey thousands of genome-wide variants

Experimental design

The authors have done an excellent job of collecting data from across the species range in New South Wales, and I commend them for this work. They have defined their objectives and identified key knowledge gaps in the literature and they've provided mostly sufficient detail of their experimental design and methods. However, I have several comments on the Methods that should be addressed:
Line 143: You say 8 degrees above but 7 degrees here. Degrees aside, I wonder if 'kilometers of coastline' might be a more meaningful measure.
Line 161: How long were they stored in CTAB buffer and at what temperature?
Line 175: I would use et al. for author list here
Line 176: What type of NGS approach did they use?
Line 188: What sort of threshold did you use for your distance matrix? Did you follow methods from another study that needs to be cited?
Line 211: Both the 'IS' and 'ST' in FIS and FST should appear as subscript.
Line 212: R packages are typically italicized in text
Line 226: 'PCs' rather than 'PC'
Line 237: Why did you choose these environmental variables? Is this all that's available from Copernicus, and if so, why not supplement with BioOracle data? Are these biologically meaningful variables? In fact, if you extracted BioOracle data you could then derive future climate variables and use this data alongside allele frequencies to calculate genomic offset across your populations.
Line 244: How did you define lack of correlation? R2 < 0.7?

Validity of the findings

I provide suggestions for the authors to consider in section 2 that I believe would strengthen the validity of their findings, but otherwise, the methods and findings are mostly robust and statistically sound. The conclusions are well stated and the authors place their work in a broader context.
Here, I outline some additional revisionary suggestions in the Results and Discussion that the authors should address:
Line 332: I understand you found very little structure in this system but it would still be useful to include the DAPC plot in the manuscript to illustrate this finding. Additionally, it would be worth calculating nucleotide diversity for each population and adding this to Table 3
Line 350: This is interesting given that most of your SNPs were associated with temp range suggesting it would be a significant predictor of genomic variation. What's your explanation for this?
Lines 356-357: I would remove this sentence and just cite Figure 4b in the following sentence.
Table 2: What do the different N columns represent in this table? It's not mentioned anywhere.
Figure 1: It would be good to overlay a raster of SST so the thermal gradient is obvious to the reader
Figure 2: I'm not entirely sure what the inset represents here, I find it very difficult to follow.
Line 386 (and elsewhere): Ensure that species names are italicized
Lines 430-432: As mentioned above, I think the authors need to be cautious in suggesting that the data presented here supports a scenario of climate change resilience. I would consider tempering this language and revising here. In theory, you could identify the most vulnerable populations by employing the genomic offset approach.
Line 459: '...single panmictic population...' is more appropriate
Lines 460-464: I think the writing could be improved here. It's possible I've misunderstood what you're saying, but how could the most northerly (warm adapted populations) be rescued by populations in the south that aren't warm-adapted?

Reviewer 3 ·

Basic reporting

The Introduction was well-researched in terms of the general impact of rapid climate change on marine organisms. However, information was lacking in terms of life-history traits for the species of interest, T. militaris e.g. generation time, reproductive strategy, larval duration, estimated (effective) population size, which provide important context for understanding the results.

For line 133: “3.9°C in mean annual sea surface temperature” I think you meant to say 3.9 C difference?

I had some suggestions for the figures and tables:
Tables
Table 2: Please explain in the main text what is the difference in the forms (spiny/incipient/smooth) if you should decide to include this column. I did not see an explanation of the different forms the main text and am curious if there were correlations between form, genetic variation, and environment.
Table 4: Please include in heading that the values in diagonal below are p-values (before/after multiple corrections, and the method used for accounting for the number of multiple comparisons.
Table 5: To enhance clarity, this table might require an additional column to show overlap between the two categories, or a statement in heading to say that there are no overlaps.

Figures
Figures should be ordered according to when they are first mentioned.
Figure 2: Please refer to Figure 2 at an appropriate location in the main text and explain what the colours and circle plot in the top left-hand corner of the figure mean. A higher resolution would be needed.
Figure 3: The figure legend could be more informative to help the reader along. It is unclear to me how the turnover functions of the curves are aggregated, and how the plots of neutral and outlier SNPs differ. I refer the authors to similar figures (Figure S1 and 2) in the supporting information of Nielsen et al. (2021) (https://doi.org/10.1111/gcb.15651)
Figure 4A: It is not clear to be which of the black dots correspond to which sampling location, text size could be increased as well.
Figure 5: Colour does not seem to be useful to me in this plot: the categories of productivity, temperature and velocity can be discussed in the main text. This plot might be better placed as a supplementary figure.

Some of the raw data headers are unfamiliar to me, are they unique to the method used? Please include explanations for the different columns in the SNP raw data document to improve data sharing.

Experimental design

This piece of research is about observing population structure in T. militaris across its latitudinal range and correlating that with patterns in connectivity and environmental variables (temperature, harvesting pressure, distance from shore and latitude) to gain insight into gene flow and standing genetic variation in the context of providing recommendations for fisheries management for this species.

With regards to the aim of observing population structure, I suggest that a STRUCTURE/admixture plot for several values of K would be a helpful supplementary visualisation, in addition to the pairwise FST plot that is already included. In addition, a population network might be useful to identify finer scale patterns of gene flow within the population, for an example please see: https://doi.org/10.1186/s12864-018-5044-8

Regarding the choice of method used for genotyping, I find it immensely more useful to the reader if the authors included a short (3-5 line summary) about how this DArTseq method works in addition to listing the references. Are the markers microsatellites? How is the genome reduction conducted? Is it possible to identify the codon position of the SNPs (if they are non-synonymous mutations)? I refer the authors to one of the numerous examples: https://doi.org/10.1371/journal.pone.0203465 on how this information can be presented.

It is unclear if the authors used a missingness threshold for SNP filtering of both datasets. There are some flags for filtering that are not typically seen e.g.: “Hamming distance threshold” are these DArt specific flags?

For the choice of physical parameters I am curious to decision behind the choice of the four variables given the many variables available, including salinity that is commonly used in seascape genomics analyses, and depth since these are benthic organisms. Please also include if these variables were collected at the surface or a specific depth.

At line 191: the authors mention that two individuals found more than 100km away were close relatives, does this correspond with their life history traits?

At line 243: Please specify which tests were used to confirm a lack of correlation between variables. This would be an important confirmation that autocorrelation in minimised in the set of variables chosen.

I am not familiar with Gradient Forest analyses and so am unable to comment on the use of the methods and results.

Validity of the findings

There were a few SNPs identified by BayPass or LFMM that are significantly correlated with environmental parameters- it would be helpful to have a list of these SNPs in addition to the complete list of SNPs already provided in the supplementary material. Are these SNPs from known regions of the genome that can be under selection?

I am also curious as to as to whether El-Nino years have an impact on the variance in the physical oceanography dataset? Can the results be discussed considering these seasonal weather patterns?

In line 405, the authors mention a “strong selection pressure” for the species. Are they able to extend this statement based on existing knowledge or from their study of the field to talk about putative drivers of selection, e.g. fishing pressure that was alluded to in the introduction?

At line 421, can the authors comment on what strengthening EAC may mean in terms of changes to connectivity and temperature?

I might challenge the assumption in line 425, does past adaptation imply future adaptation? It could be useful to think about the speed of past adaptation to environmental change and compare it to the rate of predicted future change.

Some knowledge of the life history of the provided examples would be helpful here in Line 438, does this also include those with high levels of potential gene flow from the planktotrophic larvae, and how does that relate to what we know of T. militaris?

At lines 448-449, the authors mention vulnerability to marine stochastic events, I did a very quick search and there are some related articles that would be nice to cited here e.g., Mamo et al. (2019) (https://doi.org/10.1016/j.marenvres.2019.104769); Ab Lah et al. (2018) (https://doi.org/10.1016/j.marenvres.2018.08.009) on resilience to marine heatwaves and ocean acidification.

At line 463, can the authors mention what is the expected temperature gradient from North to South for the future? And can the species migrate to track thermal shifts? With connectivity as the main driver of the population structure, how is connectivity expected to change with recent increased warming and occurrence of stochastic events and what does this mean for the resilience of the species?

Additional comments

The authors have presented their findings on the genomic population structure of T. militaris against the backdrop of connectivity and environmental parameters. As a reviewer and interested reader, I think the manuscript has a bit more scope for analyses/visualisations to increase insight from their dataset and can be better placed in context of existing research to improve understanding and recommendations for marine resource use and conservation.

---

## Round 0.2 · Minor Revisions

The authors have made an excellent job of addressing the reviewers' concerns. Two very minor aspects to address - one of which is the inclusion of the data availability statement, which appears to have been deleted by accident in the track changes version? And the authors have addressed in their response, it's just missing from the manuscript. After these, I don't think we'll need to send the manuscript back out for review - hopefully should be a quick job for you given the excellent explanation for the second point in your previous response letter.

Reviewer 1 ·

Basic reporting

No comment

Experimental design

No comment

Validity of the findings

No comment

Additional comments

I thank the authors very much for addressing my comments. The manuscript is much improved after the revisions and the authors’ responses clarified my questions. I am almost satisfied with the current version of the manuscript. I kindly ask the authors to correct the following items:

• The data availability statement is still not in the manuscript. Please make sure this appears in the final version so readers can find and access the data.
• Regarding the final selection of candidate loci used for downstream analyses: You explained very well in the response to reviewers document how you ended up with 35 unique loci. Could you add a statement like that to the manuscript please just so it is clear for the reader?

Reviewer 3 ·

Basic reporting

Thank you for uploading the data files onto dryad. The data availability statement seems to have been deleted in the pdf (I see it with strikethrough in the word version) but should be included in the final version.

Otherwise, I am satisfied with the edits to the figures that the authors have incorporated.

Experimental design

I agree that a STRUCTURE plot with k=1 would be uninformative.

I am satisfied with the additional information and amendments that the authors have provided for this section, and the reasonings that have provided in cases where they disagreed with the suggestions.

Validity of the findings

The authors have addressed my comments satisfactorily and I do not have additional suggestions to add.

---

## Round 0.3 · accepted · Accept

The authors have addressed all of the reviewers comments remaining comments. One minor thing that can be picked up in proofs - line 373: this should be Luo et al, rather than Lu et al?

The manuscript is now ready for publication. Congratulations.